# Causal Identification under Markov equivalence: Calculus, Algorithm, and Completeness

**Amin Jaber**
Purdue University
jaber0@purdue.edu

**Adèle H. Ribeiro**
Columbia University
adele@cs.columbia.edu

**Jiji Zhang**
Hong Kong Baptist University
zhangjiji@hkbu.edu.hk

**Elias Bareinboim**
Columbia University
eb@cs.columbia.edu

## Abstract

One common task in many data sciences applications is to answer questions about the effect of new interventions, like: 'what would happen to $Y$ if we make $X$ equal to $x$ while observing covariates $Z = z$?'. Formally, this is known as *conditional effect identification*, where the goal is to determine whether a post-interventional distribution is computable from the combination of an observational distribution and assumptions about the underlying domain represented by a causal diagram. A plethora of methods was developed for solving this problem, including the celebrated do-calculus [Pearl, 1995]. In practice, these results are not always applicable since they require a fully specified causal diagram as input, which is usually not available. In this paper, we assume as the input of the task a less informative structure known as a partial ancestral graph (PAG), which represents a Markov equivalence class of causal diagrams, learnable from observational data. We make the following contributions under this relaxed setting. First, we introduce a new causal calculus, which subsumes the current state-of-the-art, PAG-calculus. Second, we develop an algorithm for conditional effect identification given a PAG and prove it to be both sound and complete. In words, failure of the algorithm to identify a certain effect implies that this effect is not identifiable by any method. Third, we prove the proposed calculus to be complete for the same task.

## 1 Introduction

Despite the recent advances in AI and machine learning, the current generation of intelligent systems still lacks the pivotal ability to represent, learn, and reason with cause and effect relationships. The discipline of causal inference aims to 'algorithmitize' causal reasoning capabilities towards producing human-like machine intelligence and rational decision-making [Pearl and Mackenzie, 2018, Pearl, 2019, Bareinboim and Pearl, 2016]. One fundamental type of inference in this setting is concerned with the effect of new interventions, e.g., 'what would happen to outcome $Y$ if $X$ were set to $x$?' More generally, we may be interested in $Y$'s distribution in a sub-population picked out by the value of some covariates $Z = z$'. For example, a legislator might be interested in the impact that increasing the minimum wage ($X = x$) has on profits ($Y$) in small businesses ($Z = z$), which is written in causal language as the interventional distribution $P(y|do(x), z)$, or $P_x(y|z)$. One method capable of answering such questions is through controlled experimentation [Fisher, 1951].

In many practical settings found throughout the empirical sciences, AI, and machine learning, it is not always possible to perform a controlled experiment due to ethical, financial, and technical considerations. This motivates the study of a problem known as *causal effect identification* [Pearl,

36th Conference on Neural Information Processing Systems (NeurIPS 2022).

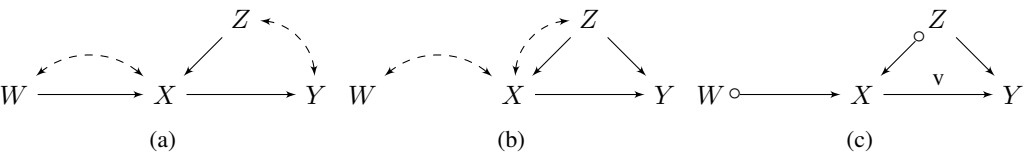

Figure 1: Sample causal diagrams (a,b) and the corresponding inferred PAG (c).

2000, Ch. 3]. The idea is to use the observational distribution $P(\mathbf{V})$ along with assumptions about the underlying domain, articulated in the form of a causal diagram $\mathcal{D}$, to infer the interventional distribution $P_{\mathbf{x}}(\mathbf{y}|\mathbf{z})$ when possible. For instance, Fig. 1a represents a causal diagram in which nodes correspond to measured variables, directed edges represent direct causal relations, and bidirected-dashed edges encode spurious associations due to unmeasured confounders. A plethora of methods have been developed to address the identification task including the celebrated causal calculus proposed by Pearl [1995] as well as complete algorithms [Tian, 2004, Shpitser and Pearl, 2006, Huang and Valtorta, 2006]. For instance, given the causal diagram in Fig. 1a and the query $P_x(y|z)$, the calculus sanctions the identity $P_x(y|z) = P(y|z, x)$. In words, the interventional distribution on the l.h.s. equates to the observational distribution on the right, which is available as input. Despite the power of these results, requiring the diagram as the input of the task is an Achilles heel for those methods, since background knowledge is usually not sufficient to pin down the single, true diagram.

To circumvent these challenges, a growing literature develops data-driven methods that attempt to learn the causal diagram from data first, and then perform identification from there. In practice, however, only an *equivalence class* (EC) of diagrams can be inferred from observational data without making substantial assumptions about the causal mechanisms [Verma, 1993, Spirtes et al., 2001, Pearl, 2000]. A prominent representation of this class is known as *partial ancestral graphs* (PAGs) [Zhang, 2008b]. Fig. 1c illustrates the PAG learned from observational data consistent with both causal diagrams in Figs. 1a and 1b since they are in the same Markov equivalence class. The directed edges in a PAG encode ancestral relations, not necessarily direct, and the circle marks stand for structural uncertainty. Directed edges labeled with $v$ signify the absence of unmeasured confounders.

Causal effect identification in a PAG is usually more challenging than from a single diagram due to the structural uncertainties and the infeasibility of enumerating each member of the EC in most cases. The do-calculus was extended for PAGs to account for the inherent structure uncertainties without the need for enumeration [Zhang, 2007]. Still, the calculus falls short of capturing all identifiable effects as we will see in Sec. 3. On the other hand, it is computationally hard to decide whether there exists (and, if so, to find) a sequence of derivations in the generalized calculus to identify an effect of interest. In a more systematic manner, a complete algorithm has been developed to identify marginal effects (i.e., $P_{\mathbf{x}}(\mathbf{y})$) given a PAG [Jaber et al., 2019a]. This algorithm can be used to identify conditional effects whenever the joint distribution $P_{\mathbf{x}}(\mathbf{y} \cup \mathbf{z})$ is identifiable. Still, many conditional effects are identifiable even if the corresponding joint effect is not (Sec. 4.2). Finally, an algorithm to identify conditional effects has been proposed in [Jaber et al., 2019b], but not proven to be complete.[1]

In this paper, we pursue a data-driven formulation for the task of identification of any conditional causal effect from a combination of an observational distribution and the corresponding PAG (instead of a fully specified causal diagram). Accordingly, we makes the following contributions:

1. We propose a causal calculus for PAGs that subsumes the stat-of-the-art calculus introduced in [Zhang, 2007]. We prove the rules are *atomic complete*, i.e., a rule is not applicable in some causal diagram in the underlying EC whenever it is not applicable given the PAG.

2. Building on these results, we develop an algorithm for the identification of conditional causal effects given a PAG. We prove the algorithm is *complete*, i.e., the effect is not identifiable in some causal diagram in the equivalence class whenever the algorithm fails.

3. Finally, we prove the calculus is *complete* for the task of identifying conditional effects.

---

[1]Another approach is based on SAT (Boolean constraint satisfaction) solvers [Hyttinen et al., 2015]. Given its somewhat distinct nature, a closer comparison lies outside the scope of this paper.

## 2 Preliminaries

In this section, we introduce the basic setup and notations. Boldface capital letters denote sets of variables, while boldface lowercase letters stand for value assignments to those variables.[2]

**Structural Causal Models.** We use Structural Causal Models (SCMs) as our basic semantical framework [Pearl, 2000]. Formally, an SCM $M$ is a 4-tuple $\langle \mathbf{U}, \mathbf{V}, \mathbf{F}, P(\mathbf{U}) \rangle$, where $\mathbf{U}$ is a set of exogenous (unmeasured) variables and $\mathbf{V}$ is a set of endogenous (measured) variables. $\mathbf{F}$ represents a collection of functions such that each endogenous variable $V_i \in \mathbf{V}$ is determined by a function $f_i \in \mathbf{F}$. Finally, $P(\mathbf{U})$ encodes the uncertainty over the exogenous variables. Every SCM is associated with one causal diagram where every variable in $\mathbf{V} \cup \mathbf{U}$ is a node, and arrows are drawn between nodes in accordance with the functions in $\mathbf{F}$. Following standard practice, we omit the exogenous nodes and add a bidirected dashed arc between two endogenous nodes if they share an exogenous parent. We only consider recursive systems, thus the corresponding diagram is acyclic. The marginal distribution induced over the endogenous variables $P(\mathbf{V})$ is called observational. The *d-separation* criterion captures the conditional independence relations entailed by a causal diagram in $P(\mathbf{V})$. For $\mathbf{C} \subseteq \mathbf{V}$, $Q[\mathbf{C}]$ denotes the post-intervention distribution of $\mathbf{C}$ under an intervention on $\mathbf{V} \setminus \mathbf{C}$, i.e. $P_{\mathbf{v} \setminus \mathbf{c}}(\mathbf{c})$.[3]

**Ancestral Graphs.** We now introduce a graphical representation of equivalence classes of causal diagrams. A MAG represents a set of causal diagrams with the same set of observed variables that entail the same conditional independence and ancestral relations among the observed variables [Richardson and Spirtes, 2002]. *M-separation* extends d-separation to MAGs such that d-separation in a causal diagram corresponds to m-separation in its unique MAG over the observed variables, and vice versa.

**Definition 1** (m-separation). *A path $p$ between $X$ and $Y$ is active (or m-connecting) relative to $\mathbf{Z}$ ($X, Y \notin \mathbf{Z}$) if every non-collider on $p$ is not in $\mathbf{Z}$, and every collider on $p$ is an ancestor of some $Z \in \mathbf{Z}$. $X$ and $Y$ are m-separated by $\mathbf{Z}$ if there is no active path between $X$ and $Y$ relative to $\mathbf{Z}$.*

Different MAGs entail the same independence model and hence are Markov equivalent. A PAG represents an equivalence class of MAGs $[\mathcal{M}]$, which shares the same adjacencies as every MAG in $[\mathcal{M}]$ and displays all and only the invariant edge marks. A circle indicates an edge mark that is not invariant. A PAG is learnable from the independence model over the observed variables, and the FCI algorithm is a standard method to learn such an object [Zhang, 2008b]. In this work, an oracle for conditional independences is assumed to be available, which leads to the true PAG.

**Graphical Notions.** Given a PAG, a path between $X$ and $Y$ is *potentially directed (causal)* from $X$ to $Y$ if there is no arrowhead on the path pointing towards $X$. $Y$ is called a *possible descendant* of $X$ and $X$ a *possible ancestor* of $Y$ if there is a potentially directed path from $X$ to $Y$. For a set of nodes $\mathbf{X}$, let $\text{An}(\mathbf{X})$ ($\text{De}(\mathbf{X})$) denote the union of $\mathbf{X}$ and the set of possible ancestors (descendants) of $\mathbf{X}$. Given two sets of nodes $\mathbf{X}$ and $\mathbf{Y}$, a path between them is called *proper* if one of the endpoints is in $\mathbf{X}$ and the other is in $\mathbf{Y}$, and no other node on the path is in $\mathbf{X}$ or $\mathbf{Y}$. Let $\langle A, B, C \rangle$ be any consecutive triple along a path $p$. $B$ is a collider on $p$ if both edges are into $B$. $B$ is a (definite) non-collider on $p$ if one of the edges is out of $B$, or both edges have circle marks at $B$ and there is no edge between $A$ and $C$. A path is *definite status* if every non-endpoint node along it is either a collider or a non-collider. If the edge marks on a path between $X$ and $Y$ are all circles, we call the path a *circle path*. We refer to the closure of nodes connected with circle paths as a *bucket*.

A directed edge $X \rightarrow Y$ in a PAG is *visible* if there exists no causal diagram in the corresponding equivalence class where the relation between $X$ and $Y$ is confounded. Which directed edges are visible is easily decidable by a graphical condition [Zhang, 2008a], so we mark visible edges by $v$.

**Manipulations in PAGs.** Let $\mathcal{P}$ denote a PAG over $\mathbf{V}$ and $\mathbf{X} \subseteq \mathbf{V}$. $\mathcal{P}_{\mathbf{X}}$ denotes the induced subgraph of $\mathcal{P}$ over $\mathbf{X}$. The $\mathbf{X}$-lower-manipulation of $\mathcal{P}$ deletes all those edges that are visible in $\mathcal{P}$ and are out of variables in $\mathbf{X}$, replaces all those edges that are out of variables in $\mathbf{X}$ but are invisible in $\mathcal{P}$ with bi-directed edges, and otherwise keeps $\mathcal{P}$ as it is. The resulting graph is denoted as $\mathcal{P}_{\underline{\mathbf{X}}}$. The $\mathbf{X}$-upper-manipulation of $\mathcal{P}$ deletes all those edges in $\mathcal{P}$ that are into variables in $\mathbf{X}$, and otherwise keeps $\mathcal{P}$ as it is. The resulting graph is denoted as $\mathcal{P}_{\overline{\mathbf{X}}}$.

---

[2]A more comprehensive discussion about the background is provided in the full report [Jaber et al., 2022].

[3]Without loss of generality, we assume the model is semi-Markovian. Tian [Tian, 2002, Sec. 5.6] shows that the identification of a causal effect in a non-Markovian model is equivalent to the identification of the same effect in a derived semi-Markovian model via a procedure known as 'projection'.

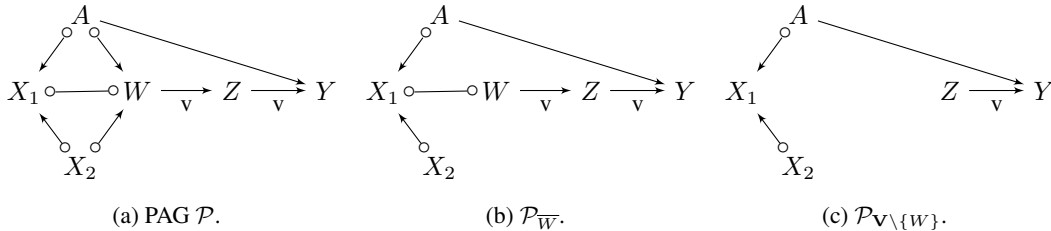

(a) PAG $\mathcal{P}$.  (b) $\mathcal{P}_{\overline{W}}$.  (c) $\mathcal{P}_{\mathbf{V} \setminus \{W\}}$.

Figure 3: Alternative methods to read ancestral relations under interventions from PAGs.

## 3 Causal Calculus for PAGs

The causal calculus introduced in [Pearl, 1995] is a seminal work that has been instrumental for understanding and eventually solving the task of effect identification from causal diagrams. Zhang [2007] generalized this result to the context of ancestral graphs, where a PAG is taken as the input of the task, instead of the specific causal diagram. In Sec. 3.1, we discuss Zhang's rules and try to understand the reasons they are insufficient to solve the identification problem in full generality. Further, in Sec.3.2, we introduce another generalization of the original calculus and prove that it is complete for atomic identification. This result will be further strengthened in subsequent sections.

### 3.1 Zhang's Calculus

An obvious extension of the m-separation criterion shown in Def. 1 to PAGs blocks all *possibly m-connecting paths*, as defined next.

**Definition 2** (Possibly m-connecting path). *In a PAG, a path $p$ between $X$ and $Y$ is a possibly m-connecting path relative to a (possibly empty) set of nodes $\mathbf{Z}$ ($X, Y \notin \mathbf{Z}$) if every definite non-collider on $p$ is not a member of $\mathbf{Z}$, and every collider on $p$ is a possible ancestor of some member of $\mathbf{Z}$. $X$ and $Y$ are $\hat{m}$-separated by $\mathbf{Z}$ if there is no possibly m-connecting path between them relative to $\mathbf{Z}$.*

Using this notion of separation, Zhang [2007] proposed a calculus given a PAG as shown next.

**Proposition 1** (Zhang's Calculus). *Let $\mathcal{P}$ be the PAG over $\mathbf{V}$, and $\mathbf{X}, \mathbf{Y}, \mathbf{W}, \mathbf{Z}$ be disjoint subsets of $\mathbf{V}$. The following rules are valid, in the sense that if the antecedent of the rule holds, then the consequent holds in every MAG and consequently every causal diagram represented by $\mathcal{P}$.*

*1.* $P(\mathbf{y}|do(\mathbf{w}), \mathbf{x}, \mathbf{z}) = P(\mathbf{y}|do(\mathbf{w}), \mathbf{z})$,  *if $\mathbf{X}$ and $\mathbf{Y}$ are $\hat{m}$-separated by $\mathbf{W} \cup \mathbf{Z}$ in $\mathcal{P}_{\overline{\mathbf{W}}}$.*

*2.* $P(\mathbf{y}|do(\mathbf{w}), do(\mathbf{x}), \mathbf{z}) = P(\mathbf{y}|do(\mathbf{w}), \mathbf{x}, \mathbf{z})$,  *if $\mathbf{X}$ and $\mathbf{Y}$ are $\hat{m}$-separated by $\mathbf{W} \cup \mathbf{Z}$ in $\mathcal{P}_{\overline{\mathbf{W}}, \underline{\mathbf{X}}}$.*

*3.* $P(\mathbf{y}|do(\mathbf{w}), do(\mathbf{x}), \mathbf{z}) = P(\mathbf{y}|do(\mathbf{w}), \mathbf{z})$,  *if $\mathbf{X}$ and $\mathbf{Y}$ are $\hat{m}$-separated by $\mathbf{W} \cup \mathbf{Z}$ in $\mathcal{P}_{\overline{\mathbf{W}}, \overline{\mathbf{X(Z)}}}$.* *where $\mathbf{X}(\mathbf{Z}) \coloneqq \mathbf{X} \setminus PossAn(\mathbf{Z})_{\mathcal{P}_{\overline{\mathbf{W}}}}$.*

In words, rule 1 generalizes m-separation to interventional settings. Further, rule 2 licenses alternating a subset $\mathbf{X}$ between intervention and conditioning. Finally, rule 3 allows the adding/removal of an intervention $\mathbf{do}(\mathbf{X} = \mathbf{x})$. The next two examples illustrate the shortcomings of this result, where the first reveals the drawback of using Def. 2 to establish graphical separation and the second inspects evaluating $\mathbf{X}(\mathbf{Z})$ in rule 3 (where the notion of possible ancestors are evoked).

**Example 1.** *Consider the PAG $\mathcal{P}$ shown in Fig. 2. Since $X$ and $Y$ are not adjacent in $\mathcal{P}$, it is easy to show that $X$ and $Y$ are separable given $\{Z_1, Z_2\}$ in every causal diagram in the equivalence class. If rule 3 of Pearl's calculus is used in each diagram, then $P_x(y|z_1, z_2) = P(y|z_1, z_2)$. Further, applying rule 2 of do-calculus in each diagram, it's also the case that $P_x(y|z_1, z_2) = P(y|z_1, z_2, x)$. However, due to the possibly m-connecting path $\langle X, Z_1, Z_2, Y \rangle$, rules 3 and 2 in Prop. 1 are not applicable to $\mathcal{P}$. In other words, even though Pearl's calculus rules 2 and 3 are applicable to each diagram in the equivalence class, the same results cannot be established by Zhang's calculus.*

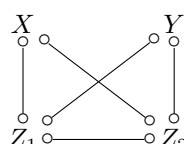

Figure 2: Sample PAG

**Example 2.** *Consider the PAG in Fig. 3a, and the evaluation on whether the equality $P_{w,x_1,x_2}(y|z) = P_w(y|z)$ holds. In order to apply rule 3 of Prop. 1, we need to evaluate whether $\{X_1, X_2\}$ is separated from $\{Y\}$ given $\{W, Z\}$ in the manipulated graph in Fig. 3b, which is not true in this case. However, the rule can be improved to be applicable in this case, as we will show later on (Sec. 3.2). The critical step will be the evaluation of the set $\mathbf{X}(\mathbf{Z})$ from $\mathcal{P}_{\overline{\mathbf{W}}}$.*

## 3.2 A New Calculus

Building on the analysis of the calculus proposed in [Zhang, 2007], we introduce next a set of rules centered around blocking *definite m-connecting paths*, as defined next.

**Definition 3** (Definite m-connecting path). *In a PAG, a path $p$ between $X$ and $Y$ is a definite m-connecting path relative to a (possibly empty) set $\mathbf{Z}$ ($X, Y \notin \mathbf{Z}$) if $p$ is definite status, every definite non-collider on $p$ is not a member of $\mathbf{Z}$, and every collider on $p$ is an ancestor of some member of $\mathbf{Z}$. $X$ and $Y$ are m-separated by $\mathbf{Z}$ if there is no definite m-connecting path between them relative to $\mathbf{Z}$.*

It is easy to see that every definite m-connecting path is a possibly m-connecting path, according to Def. 2; however, the converse is not true. For example, given the PAG in Figure 2, we have two definite status paths between $X$ and $Y$. The first is $X \circ\!\!-\!\!\circ Z_1 \circ\!\!-\!\!\circ Y$ and the second is $X \circ\!\!-\!\!\circ Z_2 \circ\!\!-\!\!\circ Y$ where $Z_1$ and $Z_2$ are definite non-colliders. Given set $\mathbf{Z} = \{Z_1, Z_2\}$, $\mathbf{Z}$ blocks all definite status paths between $X$ and $Y$. Alternatively, the path $X \circ\!\!-\!\!\circ Z_1 \circ\!\!-\!\!\circ Z_2 \circ\!\!-\!\!\circ Y$ is not definite status since $Z_1, Z_2$ are not colliders or non-colliders on this path. Hence, the path is a possibly m-connecting path relative to $\mathbf{Z}$ by Def. 2 but not a definite m-connecting path by Def. 3.

We are now ready to use this new definition and formulate a more powerful calculus.

**Theorem 1.** *Let $\mathcal{P}$ be the PAG over $\mathbf{V}$, and $\mathbf{X}, \mathbf{Y}, \mathbf{W}, \mathbf{Z}$ be disjoint subsets of $\mathbf{V}$. The following rules are valid, in the sense that if the antecedent of the rule holds, then the consequent holds in every MAG and consequently every causal diagram represented by $\mathcal{P}$.*[4]

*1. $P(\mathbf{y}|do(\mathbf{w}), \mathbf{x}, \mathbf{z}) = P(\mathbf{y}|do(\mathbf{w}), \mathbf{z})$,*      *if $\mathbf{X}$ and $\mathbf{Y}$ are m-separated by $\mathbf{W} \cup \mathbf{Z}$ in $\mathcal{P}_{\overline{\mathbf{W}}}$.*

*2. $P(\mathbf{y}|do(\mathbf{w}), do(\mathbf{x}), \mathbf{z}) = P(\mathbf{y}|do(\mathbf{w}), \mathbf{x}, \mathbf{z})$,*    *if $\mathbf{X}$ and $\mathbf{Y}$ are m-separated by $\mathbf{W} \cup \mathbf{Z}$ in $\mathcal{P}_{\overline{\mathbf{W}}, \underline{\mathbf{X}}}$.*

*3. $P(\mathbf{y}|do(\mathbf{w}), do(\mathbf{x}), \mathbf{z}) = P(\mathbf{y}|do(\mathbf{w}), \mathbf{z})$,*      *if $\mathbf{X}$ and $\mathbf{Y}$ are m-separated by $\mathbf{W} \cup \mathbf{Z}$ in $\mathcal{P}_{\overline{\mathbf{W}}, \overline{\mathbf{X}(\mathbf{Z})}}$.*
   *where $\mathbf{X}(\mathbf{Z}) := \mathbf{X} \setminus PossAn(\mathbf{Z})_{\mathcal{P}_{\mathbf{V} \setminus \mathbf{W}}}$.*

A few observations are important at this point. Despite the visual similarity to Prop. 1, there are two pivotal differences between these calculi. First, Thm. 1 only requires blocking the definite status paths, hence the use of 'm-separation' in Thm. 1 instead of '$\hat{m}$-separation'. Consider the PAG $\mathcal{P}$ in Fig. 2. We want to evaluate whether $P_x(y|z_1, z_2) = P(y|x, z_1, z_2)$ by applying Rule 2 in Theorem 1. Since all the edges in the PAG are circle edges, then $\mathcal{P}_{\underline{X}} = \mathcal{P}$. As discussed earlier, the set $\mathbf{Z} = \{Z_1, Z_2\}$ blocks all the definite status path between $X$ and $Y$. Hence, $X$ and $Y$ are m-separated by $\mathbf{Z}$ and $P_x(y|z_1, z_2) = P(y|x, z_1, z_2)$ holds true by Rule 2.

Second, Thm. 1 defines $\mathbf{X}(\mathbf{Z})$ as the subset of $\mathbf{X}$ that is not in the possible ancestors of $\mathbf{Z}$ in $\mathcal{P}_{\mathbf{V} \setminus \mathbf{W}}$, as opposed to $\mathcal{P}_{\overline{\mathbf{W}}}$ in Prop. 1. We revisit the query in Ex. 2 to clarify this subtle but significant difference. Given the PAG in Fig. 3a, we want to evaluate whether $P_{w,x_1,x_2}(y|z) = P_w(y|z)$ by applying Rule 3 from Thm. 1 instead of Prop. 1. Fig. 3c shows $\mathcal{P}_{\mathbf{V} \setminus \{W\}}$ where $\mathbf{X} = \{X_1, X_2\}$ are not possible ancestors of $Z$. Therefore, $\mathbf{X}(Z) = \mathbf{X}$, the edges into $X_1$ are cut in $\mathcal{P}_{\overline{W}, \overline{\mathbf{X}(Z)}}$, and $\mathbf{X}$ and $Y$ are m-separated therein.

Third, the proof of the Theorem 1 is provided in the appendix, but it follows from the relationship between m-connecting path in a manipulated MAG to a definite m-connecting path in the corresponding manipulated PAG. It was conjectured in [Zhang, 2008a, Footnote 15] that, for $\mathbf{X}, \mathbf{Y} \subset \mathbf{V}$, if there is an m-connecting path in $\mathcal{M}_{\overline{\mathbf{Y}}, \underline{\mathbf{X}}}$, then there is a definite m-connecting path in $\mathcal{P}_{\overline{\mathbf{Y}}, \underline{\mathbf{X}}}$. In this work, we prove that conjecture to be true for the special class of manipulations required in the rules of the calculus. Finally, the next proposition establishes the necessity of the antecedents in Thm. 1 in order to apply the corresponding rule given every diagram in the equivalence class.

---

[4]All the proofs can be found in the full report [Jaber et al., 2022].

**Theorem 2** (Atomic Completeness). *The calculus in Theorem 1 is atomically complete; meaning, whenever a rule is not applicable given a PAG, then the corresponding rule in Pearl's calculus is not applicable given some causal diagram in the Markov equivalence class.*

For instance, considering PAG $\mathcal{P}$ in Fig. 1c, we note that $(Z \not\perp\!\!\!\perp Y | X)_{\mathcal{P}_{\overline{X},\underline{Z}}}$, which means rule 2 is not applicable. Clearly, the diagram in Fig. 1a is in the equivalence class of $\mathcal{P}$ and the corresponding rule of Pearl's calculus is not applicable due to the latent confounder between $Z$ and $Y$.

## 4 Effect Identification: A Complete Algorithm

It is challenging to use the calculus rules in Thm. 1 to identify causal effects since it is computationally hard to decide whether there exists (and, if so, to find) a sequence of derivations in the generalized calculus to identify an effect of interest. The goal of this section is to formulate an algorithm to identify conditional causal effects. The next definition formalizes the notion of identifiability from a PAG, generalizing the causal-diagram-specific notion introduced in [Pearl, 2000, Tian, 2004].

**Definition 4** (Causal-Effect Identifiability). *Let $\mathbf{X}, \mathbf{Y}, \mathbf{Z}$ be disjoint sets of endogenous variables, $\mathbf{V}$. The causal effect of $\mathbf{X}$ on $\mathbf{Y}$ conditioned on $\mathbf{Z}$ is said to be identifiable from a PAG $\mathcal{P}$ if the quantity $P_{\mathbf{x}}(\mathbf{y}|\mathbf{z})$ can be computed uniquely from the observational distribution $P(\mathbf{V})$ given every causal diagram $\mathcal{D}$ in the Markov equivalence class represented by $\mathcal{P}$.*

The remainder of the section is organized as follows. Sec. 4.1 introduces a version of the **IDP** algorithm [Jaber et al., 2019a] to identify marginal causal effects. The attractiveness of this version is that it yields simpler expressions whenever the effect is identifiable while preserving the same expressive power, i.e., completeness for marginal identification. Sec. 4.2 utilizes the new algorithm along with the calculus in Thm. 1 to formulate a complete algorithm for conditional identification.

### 4.1 Marginal Effect Identification

We introduce the notion of pc-component next, which generalizes the notion of c-component that is instrumental to solve identification problems in a causal diagram [Tian and Pearl, 2002].

**Definition 5** (PC-Component). *In a PAG, or any induced subgraph thereof, two nodes are in the same possible c-component (pc-component) if there is a path between them such that (1) all non-endpoint nodes along the path are colliders, and (2) none of the edges is visible.*

Following Def. 5, e.g., $W$ and $Z$ in Fig. 1c are in the same pc-component due to $W \circ\!\!\rightarrow X \leftarrow\!\!\circ Z$. By contrast, $X, Y$ are not in the same pc-component since the direct edge between them is visible and $Z$ along $\langle X, Z, Y \rangle$ is not a collider. Building on pc-components, we define the key notion of regions.

**Definition 6** (Region $\mathcal{R}_{\mathbf{A}}^{\mathbf{C}}$). *Given PAG $\mathcal{P}$ over $\mathbf{V}$, and $\mathbf{A} \subseteq \mathbf{C} \subseteq \mathbf{V}$. The region of $\mathbf{A}$ with respect to $\mathbf{C}$, denoted $\mathcal{R}_{\mathbf{A}}^{\mathbf{C}}$, is the union of the buckets that contain nodes in the pc-component of $\mathbf{A}$ in $\mathcal{P}_{\mathbf{C}}$.*

A region expands a pc-component and will prove to be useful in the identification algorithm. For example, the pc-component of $X$ in Fig. 2 is $\{X, Z_1, Z_2\}$ and the region $\mathcal{R}_X^{\mathbf{V}} = \{X, Z_1, Z_2, Y\}$. Building further on these definitions and the new calculus, we derive a new identification criterion.

**Proposition 2.** *Let $\mathcal{P}$ denote a PAG over $\mathbf{V}$, $\mathbf{T}$ be a union of a subset of the buckets in $\mathcal{P}$, and $\mathbf{X} \subset \mathbf{T}$ be a bucket. Given $P_{\mathbf{v}\setminus\mathbf{t}}$ (i.e., an observational expression for $Q[\mathbf{T}]$), $Q[\mathbf{T} \setminus \mathbf{X}]$ is identifiable by the following expression if, in $\mathcal{P}_{\mathbf{T}}$, $C^{\mathbf{X}} \cap PossDe(\mathbf{X}) \subseteq \mathbf{X}$, where $C^{\mathbf{X}}$ is the pc-component of $\mathbf{X}$.*

$$Q[\mathbf{T} \setminus \mathbf{X}] = \frac{P_{\mathbf{v}\setminus\mathbf{t}}}{P_{\mathbf{v}\setminus\mathbf{t}}(\mathbf{X}|\mathbf{T} \setminus PossDe(\mathbf{X}))} \tag{1}$$

Note the interventions are over buckets which may or may not be single nodes. Since there is little to no causal information inside a bucket, marginal effects of interventions over subsets of buckets are not identifiable. Also, the input distribution is possibly interventional which licenses recursive applications of the criterion. The next example illustrates the power of the new criterion.

**Example 3.** *Consider PAG $\mathcal{P}$ in Fig. 3a and the query $P_{x_1,x_2,w}(y, z, a)$. Starting with the observational distribution $P(\mathbf{V})$ as input, let $\mathbf{T} = \mathbf{V}$ and $\mathbf{X} = \{X_1, W\}$. We have $C^{\mathbf{X}} = \{X_1, W, A, X_2\}$,*

**Algorithm 1** IDP($\mathcal{P}, \mathbf{x}, \mathbf{y}$)

> **Input:** PAG $\mathcal{P}$ and two disjoint sets $\mathbf{X}, \mathbf{Y} \subset \mathbf{V}$
> **Output:** Expression for $P_\mathbf{x}(\mathbf{y})$ or FAIL

1: Let $\mathbf{D} = \texttt{PossAn}(\mathbf{Y})_{\mathcal{P}_{\mathbf{V} \setminus \mathbf{x}}}$
2: **return** $\sum_{\mathbf{d} \setminus \mathbf{y}}$ IDENTIFY($\mathbf{D}, \mathbf{V}, P$)

3: **function** IDENTIFY($\mathbf{C}, \mathbf{T}, Q = Q[\mathbf{T}]$)
4:      **if** $\mathbf{C} = \emptyset$ **then return** 1
5:      **if** $\mathbf{C} = \mathbf{T}$ **then return** $Q$

     /* In $\mathcal{P}_\mathbf{T}$, let $\mathbf{B}$ denote a bucket, and let $C^\mathbf{B}$ denote the pc-component of $\mathbf{B}$ */
6:      **if** $\exists \mathbf{B} \subset \mathbf{T} \setminus \mathbf{C}$ such that $C^\mathbf{B} \cap \texttt{PossDe}(\mathbf{B})_{\mathcal{P}_\mathbf{T}} \subseteq \mathbf{B}$ **then**
7:          Compute $Q[\mathbf{T} \setminus \mathbf{B}]$ from $Q$;                      $\triangleright$ via Prop. 2
8:          **return** IDENTIFY($\mathbf{C}, \mathbf{T} \setminus \mathbf{B}, Q[\mathbf{T} \setminus \mathbf{B}]$)
9:      **else if** $\exists \mathbf{B} \subset \mathbf{C}$ such that $\mathcal{R}_\mathbf{B} \neq \mathbf{C}$ **then**          $\triangleright$ $\mathcal{R}_\mathbf{B}$ is equivalent to $\mathcal{R}^\mathbf{C}_\mathbf{B}$
10:          **return** $\dfrac{\text{IDENTIFY}(\mathcal{R}_\mathbf{B}, \mathbf{T}, Q) \times \text{IDENTIFY}(\mathcal{R}_{\mathbf{C} \setminus \mathcal{R}_\mathbf{B}}, \mathbf{T}, Q)}{\text{IDENTIFY}(\mathcal{R}_\mathbf{B} \cap \mathcal{R}_{\mathbf{C} \setminus \mathcal{R}_\mathbf{B}}, \mathbf{T}, Q)}$
11:      **else throw** FAIL

$\texttt{PossDe}(\mathbf{X}) = \{X_1, W, Z, Y\}$, and $C^\mathbf{X} \cap \texttt{PossDe}(\mathbf{X}) = \mathbf{X}$. *Hence, the criterion in Prop. 2 is applicable and we have* $P_\mathbf{x}(y, z, a, x_2) = \frac{P(\mathbf{v})}{P(x_1, w | a, x_2)} = P(a, x_2) \times P(y, z | a, w)$ *after simplification. Next, we consider intervening on $X_2$ given* $P_{x_1, w}(y, z, a, x_2)$. *Notice $X_2$ is disconnected from the other nodes in $\mathcal{P}_{\mathbf{V} \setminus \{X_1, W\}}$ and it trivially satisfies the criterion in Prop. 2. Therefore, we get the expression* $P_{x_1, x_2, w}(y, z, a) = \frac{P_{x_1, w}(y, z, a, x_2)}{P_{x_1, w}(x_2 | y, z, a)} = P(a) \times P(y, z | w, a)$ *after simplification.*

A more general criterion was introduced in [Jaber et al., 2018, Thm. 1] based on the *possible children* of the intervention bucket $\mathbf{X}$ instead of the possible descendants. However, the corresponding expression is convoluted and usually large, which could be intractable even if the effect is identifiable. Alg. 1 shows the proposed version of **IDP**, which builds on the new criterion (Prop. 2). Specifically, the key difference between this algorithm and the one proposed in [Jaber et al., 2019a] is in Lines 6-7, where the criterion in Prop. 2 is used as opposed to that in [Jaber et al., 2018, Thm. 1]. Interestingly enough, the new criterion is "just right," namely, it is also sufficient to obtain a complete algorithm for marginal effect identification, as shown in the next result.[5]

**Theorem 3** (completeness). *Alg. 1 is complete for identifying marginal effects $P_\mathbf{x}(\mathbf{y})$. Moreover, the calculus in Thm. 1, together with standard probability manipulations are complete for the same task.*

### 4.2 Conditional Effect Identification

We start by making a couple of observations, and then build on those observations to formulate an algorithm to identify conditional causal effects. The proposed algorithm leverages the calculus in Thm. 1 and the **IDP** algorithm in Alg. 1. Obs. 1 notes that a conditional effect $P_\mathbf{x}(\mathbf{y}|\mathbf{z})$ can be rewritten as $\frac{P_\mathbf{x}(\mathbf{y}, \mathbf{z})}{\sum_{\mathbf{y}'} P_\mathbf{x}(\mathbf{y}', \mathbf{z})}$, and hence it is identifiable if $P_\mathbf{x}(\mathbf{y}, \mathbf{z})$ is identifiable by Alg. 1.

**Observation 1** (Marginal Effect). *Consider PAG $\mathcal{P}_1$ in Fig. 4a where the goal is to identify the causal effect $P_b(a, c|d)$. We notice that the effect $P_b(a, c, d)$ is identifiable using the **IDP** algorithm. Let $E := P(a, d) \times P(c|b, d)$ denote the expression for the marginal effect $P_b(a, c, d)$ which can be obtained from **IDP**. Consequently, the target effect can be computed using the expression $E / \sum_{a', c'} E$.*

Whenever the marginal effect $P_\mathbf{x}(\mathbf{y}, \mathbf{z})$ is not identifiable using Alg. 1, Observations 2 and 3 propose techniques to identify the conditional effect using the calculus in Thm. 1, namely rule 2. Obs. 2 uses rule 2 of Thm. 1, when applicable, to move variables from the conditioning to the intervention set. The marginal effect of the resulting conditional query turns out to be identifiable, and consequently does the conditional effect. We note that the work in [Shpitser and Pearl, 2006] uses the same trick to formulate an algorithm for conditional effect identification given a causal diagram.

---

[5]A more detailed comparison of the two algorithms along with illustrative examples is provided in the full report [Jaber et al., 2022].

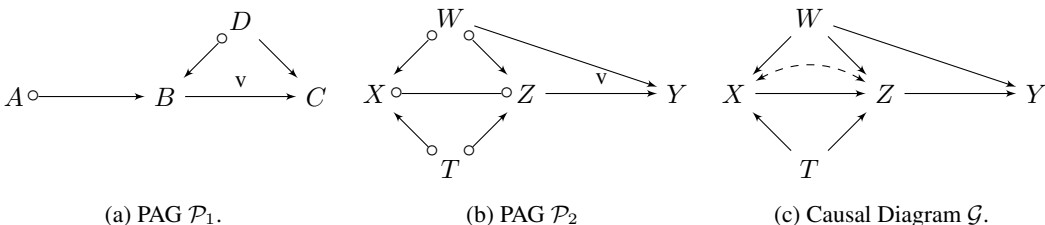

(a) PAG $\mathcal{P}_1$.        (b) PAG $\mathcal{P}_2$        (c) Causal Diagram $\mathcal{G}$.

Figure 4: (a,b) Two sample PAGs, and (c) a causal diagram in the equivalence class of (b).

---

**Algorithm 2** CIDP$(\mathcal{P}, \mathbf{x}, \mathbf{y}, \mathbf{z})$

    **Input:** PAG $\mathcal{P}$ and three disjoint sets $\mathbf{X}, \mathbf{Y}, \mathbf{Z} \subset \mathbf{V}$
    **Output:** Expression for $P_{\mathbf{x}}(\mathbf{y}|\mathbf{z})$ or FAIL

1:  $\mathbf{D} \leftarrow \text{PossAn}(\mathbf{Y} \cup \mathbf{Z})_{\mathcal{P}_{\mathbf{V} \setminus \mathbf{x}}}$
    /* Let $\mathbf{B}_1, \ldots, \mathbf{B}_m$ denote the buckets in $\mathcal{P}$ */
2: **while** $\exists \mathbf{B}_i$ s.t. $\mathbf{B}_i \cap \mathbf{D} \neq \emptyset \wedge \mathbf{B}_i \nsubseteq \mathbf{D}$ **do**
3:     $\mathbf{X}' \leftarrow \mathbf{B}_i \cap \mathbf{X}$
4:     **if** $(\mathbf{X}' \perp\!\!\!\perp \mathbf{Y}|(\mathbf{X} \setminus \mathbf{X}') \cup \mathbf{Z})_{\mathcal{P}_{\overline{\mathbf{X} \setminus \mathbf{X}'}, \underline{\mathbf{x}'}}}$ **then**
5:         $\mathbf{x} \leftarrow \mathbf{x} \setminus \mathbf{x}'; \mathbf{z} \leftarrow \mathbf{z} \cup \mathbf{x}'$            ▷ Apply rule 2 of Thm. 1
6:         $\mathbf{D} \leftarrow \text{PossAn}(\mathbf{Y} \cup \mathbf{Z})_{\mathcal{P}_{\mathbf{V} \setminus \mathbf{x}}}$
7:     **else throw** FAIL

    /* Let $\mathbf{Z}_1, \ldots, \mathbf{Z}_m$ partition $\mathbf{Z}$ such that $\mathbf{Z}_i := \mathbf{Z} \cap \mathbf{B}_i$ */
8: **while** $\exists \mathbf{Z}_i$ s.t. $(\mathbf{Z}_i \perp\!\!\!\perp \mathbf{Y}|\mathbf{X} \cup (\mathbf{Z} \setminus \mathbf{Z}_i))_{\mathcal{P}_{\overline{\mathbf{x}}, \underline{\mathbf{z}_i}}}$ **do**
9:     $\mathbf{x} \leftarrow \mathbf{x} \cup \mathbf{z}_i; \mathbf{z} \leftarrow \mathbf{z} \setminus \mathbf{z}_i$            ▷ Apply rule 2 of Thm. 1

10: $E \leftarrow \textbf{IDP}(\mathcal{P}, \mathbf{x}, \mathbf{y} \cup \mathbf{z})$
11: **return** $E / \sum_{\mathbf{y}'} E$

---

**Observation 2** (Flip Observations to Interventions). *Consider PAG $\mathcal{P}_1$ in Fig. 4a and the causal query $P_a(c|b,d)$. Unlike the case in Obs. 1, the marginal effect $P_a(b,c,d)$ is not identifiable by the* **IDP** *algorithm. Using rule 2 of Thm. 1, we have $(B \perp\!\!\!\perp C|D)_{\mathcal{P}_{\overline{A}, \underline{B}}}$ and we move B from conditioning to intervention, i.e., $P_a(c|b,d) = P_{a,b}(c|d)$. The marginal effect $P_{a,b}(c,d)$ is identifiable by* **IDP** *and we get the expression $E := P(d) \times P(c|b,d)$. Hence, we have $P_a(c|b,d) = P_{a,b}(c|d) = E / \sum_{c'} E$.*

Finally, Obs. 3 comes as a surprise since it requires flipping interventions to observations, contrary to Obs. 2. A key graphical structure in the PAG that requires such a treatment is the presence of a proper possibly directed path from $\mathbf{X}$ to $\mathbf{Y} \cup \mathbf{Z}$ that starts with a circle edge.

**Observation 3** (Flip Interventions to Observations). *We revisit the query in Example. 1. First, the marginal effect $P_x(y, z_1, z_2)$ is not identifiable by the* **IDP** *algorithm. Also, we cannot use rule 2 in Thm. 1 to flip $Z_1$ or $Z_2$ into interventions since they are both adjacent to Y with a circle edge (○─○). However, we can use rule 2 to flip X to the conditioning set since there are no definite m-connecting paths between X and Y given $\{Z_1, Z_2\}$ in the PAG. Hence, we obtain $P_x(y|z_1, z_2) = P(y|z_1, z_2, x)$. Alternatively, consider PAG $\mathcal{P}_2$ in Fig. 4b with the causal query $P_x(y|z)$. We cannot use rule 2 to flip X to an observation since $\langle X, W, Y \rangle$ is active given Z in $\mathcal{P}_{2X}$. In fact, the causal diagram $\mathcal{G}$ in Fig. 4c is in the equivalence class of $\mathcal{P}_2$ and such that $P_x(y|z)$ is provably not identifiable [Shpitser and Pearl, 2006, Corol. 2]. Hence, the effect is not identifiable given $\mathcal{P}_2$ according to Def. 4.*

Putting these observations together, we formulate the **CIDP** algorithm (Alg. 2) for identifying conditional causal effects given a PAG. The algorithm is divided into three phases. In Phase I (lines 1-7), Obs. 3 is used to check for proper possibly directed paths from $\mathbf{X}$ to $\mathbf{Y} \cup \mathbf{Z}$ that start with a circle edge. This is checked algorithmically by computing $\mathbf{D} = \text{PossAn}(\mathbf{Y} \cup \mathbf{Z})_{\mathcal{P}_{\mathbf{V} \setminus \mathbf{x}}}$, iteratively, and checking if some bucket $\mathbf{B}_i$ in $\mathcal{P}$ intersects with, but is not a subset of, $\mathbf{D}$. If such a bucket exists, **CIDP** flips $\mathbf{B}_i \cap \mathbf{X}$ from interventions to observations using rule 2, when applicable, else the algorithm throws a fail and the effect is not computable. In Phase II (lines 8-9), Obs. 2 is used to flip the subset of observations in each bucket into interventions by applying rule 2 of Thm. 1,

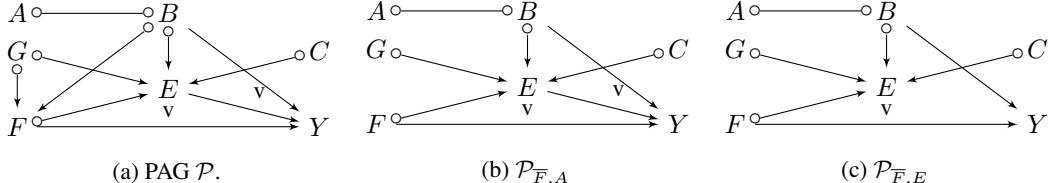

(a) PAG $\mathcal{P}$.         (b) $\mathcal{P}_{\overline{F},\underline{A}}$         (c) $\mathcal{P}_{\overline{F},\underline{E}}$

Figure 5: Illustrating example for Alg. 2.

whenever applicable. Finally, in Phase III (line 10), the marginal effect $P_{\mathbf{x}}(\mathbf{y} \cup \mathbf{z})$ is computed from the modified sets $\mathbf{X}$ and $\mathbf{Z}$, using the **IDP** algorithm in Alg. 1. If the call is successful, an expression for the conditional effect is returned at line 11. The example below illustrates **CIDP** in action. An empirical evaluation of **CIDP** is provided in the full report [Jaber et al., 2022].[6]

**Example 4.** *Consider PAG $\mathcal{P}$ in Fig. 5a and the conditional query $P_{\mathbf{x}}(\mathbf{y}|\mathbf{z}) \coloneqq P_{a,f}(y|b,e)$. In Phase I, we have $\mathbf{D} = \mathit{PossAn}(\mathbf{Y} \cup \mathbf{Z})_{\mathcal{P}_{\mathbf{V}\setminus\mathbf{x}}} = \{Y, B, E, C, G\}$, and the bucket $\{A, B\}$ satisfies the conditions at line 2 since $A \notin \mathbf{D}$. In $\mathcal{P}_{\overline{F},\underline{A}}$ (as shown in Fig. 5b), $\mathbf{X}' = \{A\}$ is m-separated from $Y$ given $\{B, E, F\}$ which satisfies the if condition at line 4. Hence, we flip $A$ to the conditioning set $\mathbf{Z}$ via rule 2 of Thm. 1 to obtain the updated query $P_{\mathbf{x}}(\mathbf{y}|\mathbf{z}) = P_f(y|a, b, e)$. In Phase II (lines 8-9), let $\mathbf{Z}_1 = \{E\}$ and $\mathbf{Z}_2 = \{A, B\}$. In $\mathcal{P}_{\overline{F},\underline{E}}$ (see Fig. 5c), we have $E$ m-separated from $Y$ given $\{F\} \cup \mathbf{Z}_2$ which satisfies the if condition at line 8. Hence, we flip $E$ to the intervention set using rule 2 of Thm. 1 and we get the updated query $P_{\mathbf{x}}(\mathbf{y}|\mathbf{z}) = P_{e,f}(y|a, b)$. Next, we check if $\mathbf{Z} = \mathbf{Z}_2$ is m-separated from $Y$ given $\mathbf{X}$ in $\mathcal{P}_{\overline{\mathbf{X}},\underline{\mathbf{Z}}}$ which does not hold due to a bidirected edge between $B$ and $Y$. Hence, rule 2 is not applicable and $\mathbf{Z}$ remains in the conditioning set. Finally, we call **IDP** to compute the marginal effect $P_{e,f}(y, a, b)$, if possible. The effect is identifiable with the simplified expression $P(y|b, e, f) \times P(a, b)$. Hence, $P_{a,f}(y|b, e) = P_{e,f}(y|a, b) = \frac{P(y|b,e,f) \times P(a,b)}{\sum_{y'} P(y'|b,e,f) \times P(a,b)} = P(y|b, e, f)$.*

The soundness of Alg. 2 follows from that of Alg. 1 and Thm. 1. Next, we turn to its completeness. According to Def. 4, whenever **CIDP** fails, we need to establish one of two conditions for completeness. Either there exist two causal diagrams in the equivalence class with different identifications, or the effect is not identifiable in some causal diagram according to the criterion in [Shpitser and Pearl, 2006, Corol. 2]. Thm. 4 establishes completeness by proving that the latter is always the case. This result along with the completeness of the calculus rules for the identification of marginal effects (see Thm. 3) implies that the rules are complete for conditional effects as well.

**Theorem 4** (completeness). *Alg. 2 is complete for identifying conditional effects $P_{\mathbf{x}}(\mathbf{y}|\mathbf{z})$. Also, the calculus in Thm. 1, together with standard probability manipulations are complete for the same task.*

## 5 Discussion

In this work, an oracle for conditional independences is assumed to be available, which leads to the true PAG. Assuming the presence of an oracle for conditional independence encapsulates the challenge of dealing with finite data and of testing for conditional independence thereof. Another challenge lies in the computational complexity of learning the PAG in the first place [Colombo et al., 2012], and estimating the expression when the effect is identifiable [Pearl and Robins, 1995, Jung et al., 2021]. In light of this, it is important to make the distinction between the task of causal effect identification and that of causal effect estimation.

This set of results is concerned with the first task (causal identification), which asks whether a target conditional effect is uniquely computable from $P(\mathbf{V})$, the observational distribution, and given a PAG learnable from $P(\mathbf{V})$. The objective of **CIDP** in Algorithm 2 is to decide whether the effect is identifiable and provide an expression for it when the answer is yes, while being agnostic as to whether $P(\mathbf{V})$ can be accurately estimated from the available samples.

As for the second task, the estimation of the conditional causal effect using the identification formula provided by Algorithm 2 poses several challenges under finite data. The number of samples sufficient to identify a given effect would depend on the size of the expression, among other factors, and

---

[6]Code is available at `https://github.com/CausalAILab/PAGId`

naive methods for estimation exacerbate this problem. Recent work such as [Jung et al., 2021] proposes a double machine learning estimator for marginal effects that are identifiable given a PAG. An interesting direction of work is to generalize this approach to conditional causal effects that are identifiable by **CIDP**.

## 6 Conclusions

In this work, we investigate the problem of identifying conditional interventional distributions given a Markov equivalence class of causal diagrams represented by a PAG. We introduce a new generalization of the do-calculus for identification of interventional distributions in PAGs (Thm. 1) and show it to be atomically complete (Thm. 2). Building on these results, we develop the **CIDP** algorithm (Alg. 2), which is both sound and complete, i.e., it identifies any conditional effects of the form $P_{\mathbf{x}}(\mathbf{y}|\mathbf{z})$ that is identifiable (Thm. 4). Finally, we show that the new calculus rules, along with standard probability manipulations, are complete for the same task. These results close the problem of effect identification under Markov equivalence in that they completely delineate the theoretical boundaries of what is, in principle, computable from a certain data collection. We expect the newly introduced machinery to help data scientists to identify novel effects in real world settings.

## Acknowledgments and Disclosure of Funding

Bareinboim and Ribeiro's research was supported in part by the NSF, ONR, AFOSR, DoE, Amazon, JP Morgan, and The Alfred P. Sloan Foundation. Zhang's research was supported by the RGC of Hong Kong under GRF13602720.

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
