# OpenReview forum: "Causal Identification under Markov equivalence: Calculus, Algorithm, and Completeness"
_NeurIPS.cc/2022/Conference — NeurIPS 2022 Accept_

### Official Review · Reviewer_9ZqV · 2022-06-26

**Rating:** 9
**Confidence:** 4
**Soundness:** 4 excellent
**Presentation:** 4 excellent
**Contribution:** 4 excellent

**Summary:**

The manuscript provides a sound and complete calculus and algorithm for identifying conditional causal effects of the form $p(\bf{y} | do(\bf{x}), \bf{z})$ in partial ancestral graphs (PAGs), i.e. a Markov equivalence class of maximal ancestral graphs (MAGs). This generalizes prior work on causal effect identification in different kinds of graphs and equivalence classes thereof.

**Questions:**

See above.

**Limitations:**

I am not convinced that the brief reference to an independence oracle constitutes a thorough discussion of the method’s limitations. Some elaboration would be welcome here, as well as further comments on if/how the method can be used for counterfactual effect identification, computational efficiency, etc.

**Strengths And Weaknesses:**

Completeness results for conditional effect identification in PAGs are notably lacking, despite strong progress in this area by Zhang (2007) and Jaber et al. (2019a, 2019b). Providing such a result – not just for an abstract set of rules but in an efficient algorithm – constitutes a major contribution. Since the FCI algorithm outputs a PAG, CIPD could plausibly be incorporated into a causal discovery pipeline for effect estimation over a learned Markov equivalence class. This could be of great use to practitioners in a variety of fields. The manuscript is well-researched and well-written, quite clear and easy to follow despite considerable technicalities. I commend the authors for their strong and original work.

If anything, I wonder if a conference submission is really the right venue for this, given how many follow up questions I have! I fully appreciate that space constraints prevent the authors from tackling all the problems that arise in causal reasoning with PAGs, but I strongly encourage them to consider a journal length follow up article, along the lines of Zhang (2007) or Shpitser & Pearl (2008). Short of that, some revisions to the present manuscript at least addressing these concerns would be welcome:

-The do-calculus is famously complete for effect identification across all three levels of the causal hierarchy (Shpitser & Pearl, 2008). How, if at all, could the methods proposed here be leveraged for counterfactual inference in PAGs? While a complete answer may lie beyond the scope of this manuscript, some discussion of the problem would be interesting, if only as a direction for future work.

-There is some discussion in the manuscript of an R implementation. Mentioning this without providing any empirical results or supplemental code is a bit frustrating, as it’s unclear how reviewers are meant to evaluate the performance or usability of the CIDP algorithm. I would either include this material as part of the submission or cut the reference altogether.

-There has been considerable work in recent years on CATE estimation. Could methods like inverse propensity weights or double machine learning be used to estimate conditional causal effects in PAGs, at least under certain conditions? Again, perhaps beyond scope to go too deep into this, but a natural follow up that will likely occur to many readers.

---

> ### Author Response · Authors · 2022-08-02
> **Response to Reviewer 9ZqV (Part 2/2)**
>
> 4. “I am not convinced that the brief reference to an independence oracle constitutes a thorough discussion of the method’s limitations.”
>
> Assuming the presence of an oracle for conditional independence encapsulates the challenge of dealing with finite data and of testing for conditional independence thereof. Another challenge lies in the computational complexity of learning the PAG in the first place (Colombo et al., 2012), and estimating the expression when the effect is identifiable (Jung et al., 2021). We will elaborate on these issues in the manuscript. Thanks for the excellent point.
>
> (Bareinboim et al., 2022) Elias Bareinboim, Juan D. Correa, Duligur Ibeling, and Thomas Icard. “On pearl’s hierarchy and the foundations of causal inference.” In Probabilistic and Causal Inference: The Works of Judea Pearl, pp. 507-556. 2022.
>
> (Jung et al., 2021) Yonghan Jung, Jin Tian, and Elias Bareinboim. “Estimating identifiable causal effects on Markov equivalence class through double machine learning.” International Conference on Machine Learning. PMLR, 2021.
>
> (Correa et al. 2021) Juan D. Correa, Sanghack Lee, and Elias Bareinboim. "Nested counterfactual identification from arbitrary surrogate experiments." Advances in Neural Information Processing Systems 34 (2021): 6856-6867.
>
> (Lee et al., 2019) Sanghack Lee, Juan D. Correa, and Elias Bareinboim. "General identifiability with arbitrary surrogate experiments." In Uncertainty in artificial intelligence, pp. 389-398. PMLR, 2020.
>
> (Shpitser and Sherman, 2018) Ilya Shpitser and Eli Sherman. "Identification of personalized effects associated with causal pathways." In Proceedings of the Conference on Uncertainty in Artificial Intelligence, 2018.
>
> (Zhang and Bareinboim, 2018) Junzhe Zhang and Elias Bareinboim. “Fairness in decision-making—the causal explanation formula.” In Proceedings of the AAAI Conference on Artificial Intelligence, vol. 32, no. 1. 2018.
>
> (Chernozhukov et al., 2018) Victor Chernozhukov, Denis Chetverikov, Mert Demirer, Esther Duflo, Christian Hansen, Whitney Newey, James Robins. “Double/debiased machine learning for treatment and structural parameters.” The Econometrics Journal, 21(1), 2018.
>
> (Colombo et al., 2012) Diego Colombo, Marloes H. Maathuis, Markus Kalisch, and Thomas S. Richardson. "Learning high-dimensional directed acyclic graphs with latent and selection variables." The Annals of Statistics (2012): 294-321.
>
> (Shpitser and Pearl, 2008) Ilya Shpitser and Judea Pearl. “Complete identification methods for the causal hierarchy.” Journal of Machine Learning Research 9 (2008): 1941-1979.
>
> (Avin et al., 2005) Chen Avin, Ilya Shpitser, and Judea Pearl. "Identifiability of path-specific effects." In Proceedings of the 19th international joint conference on Artificial intelligence, pp. 357-363. 2005.
>
> (Pearl, 2001) Judea Pearl. "Direct and indirect effects." In Proceedings of the 17th Conference on Uncertainty in Artificial Intelligence, 2001.
>
> (Halpern, 2000) Joseph Y. Halpern. "Axiomatizing causal reasoning." Journal of Artificial Intelligence Research 12 (2000): 317-337.

---

> ### Author Response · Authors · 2022-08-02
> **Response to Reviewer 9ZqV (Part 1/2)**
>
> Thank you for the very favorable assessment and all the feedback provided. We are surely considering your suggestion of a journal submission to provide a unified and thorough treatment of the identification problem in PAGs. Below, we address the issues you raised.
>
> 1. "The do-calculus is famously complete for effect identification across all three levels of the causal hierarchy (Shpitser & Pearl, 2008). How, if at all, could the methods proposed here be leveraged for counterfactual inference in PAGs?"
>
> Thank you for the good and non-trivial question. Perhaps there are different ways of thinking about this, and we would like to share two that we consider complementaries.
>
> First, the do-calculus is specific for queries in layer 2, so the name of the calculus. This means that it can express counterfactuals of the form $P(Y_x | Z_x)$, for when the subscript of all terms are the same; also, it doesn’t allow for nesting. In other words, quantities such as $P(Y_x | Z_{x’})$, where $x \neq x’$ are not allowed in the do-language and in layer 2. For a more technical discussion on the semantics of this and layer 2 queries, Def. 5/Eq. 1.9 in (Bareinboim et al., 2022) could be enlightening. Having conflicting subscripts (worlds) is a distinct feature of layer 3. Having said that, it’s not the case that the do-calculus solves any counterfactual query in layer 3, but just do-queries related to layers 1 and 2. The current most general result in terms of the do-calculus we believe is (Corollary 2, Lee et al., 2019), which may also contain a good survey.
>
> Second, the work you mentioned – (Shpitser and Pearl, 2008) – provides methods to perform counterfactual reasoning in many settings but not in all, including as discussed in (Pearl, 2001, Avin et al., 2005). Examples of more recent work relaxing some of the conditions of this work and prior literature include (Shpitser and Sherman, 2018; Zhang and Bareinboim, 2018), and a more recent, claimed complete method for arbitrary nested counterfactual identification in (Correa et al., 2021). It’s out of the scope of this paper to do a full survey on the literature on layer 3 identification, so we apologize beforehand if we missed some references or nuance. To be clear, we make no claim about counterfactual identification and don’t feel this result follows immediately from our or these other works in the literature. One thing that is interesting in these works, as we understand them, is that they do not use calculus per se, and follow a more algorithmic approach, generalizing Tian’s ID/C-component algorithm. On the other, more logical side, (Halpern, 2000) has his own axiomatization, and even though is more focused on interventional quantities, it could be used to think about counterfactuals.
>
> One distinctive challenge with counterfactual queries in the context of PAGs is how to generalize the notion of counterfactual factorization and graphs so as to read off relevant properties from it. We do feel the results in our paper can help provide some insights on this challenge, but it’s incomplete and a current, ongoing, and challenging project. We will add some remarks on this, thank you for raising the issue.
>
> 2. “There is some discussion in the manuscript of an R implementation. Mentioning this without providing any empirical results or supplemental code is a bit frustrating, as it’s unclear how reviewers are meant to evaluate the performance or usability of the CIDP algorithm. I would either include this material as part of the submission or cut the reference altogether.”
>
> An empirical evaluation of CIDP is provided in the current supplementary material, please, see Appx. G. Unfortunately, we had to move it to the appendix due to space limitations and to ensure a thorough and clear presentation of the theoretical claims. We plan to utilize the extra page in the final version, if the paper is accepted, to present the empirical evaluation. We are currently polishing the code before releasing it to the public.
>
> 3. “There has been considerable work in recent years on CATE estimation. Could methods like inverse propensity weights or double machine learning be used to estimate conditional causal effects in PAGs, at least under certain conditions? Again, perhaps beyond scope to go too deep into this, but a natural follow up that will likely occur to many readers.”
>
> This is a good suggestion indeed. Recent work utilizes frameworks such as double machine learning to estimate marginal causal effects given a PAG (Jung et al., 2021), following the work of (Chernozhukov et al., 2018). A natural future research direction is to use similar techniques to estimate conditional causal effects as well.

---

### Official Review · Reviewer_CdvZ · 2022-07-11

**Rating:** 9
**Confidence:** 3
**Soundness:** 4 excellent
**Presentation:** 3 good
**Contribution:** 4 excellent

**Summary:**

This is a paper about the identification of causal queries in structural causal models. If most of the existing works assume the availability of the causal graph, here only the PAG (partial ancestral graph) structure (corresponding to a collection of CGs) is assumed to be available. The authors derive an analogous of do-calculus for such setup. This is an extension of Zhang's calculus, but here completeness guarantees are provided. Algorithms for the identification of causal effects are provided.

**Questions:**

- It is not perfectly clear to me whether or not these results can be also applied to counterfactual queries (I think so but this point might be made explicit).
- The considered approach allows to address identifiability when the causal graph is not known and only the PAG is available. As I understand, PAG identifiability corresponds to identifiability on all the CGs compatible with the PAG and the fact that the query would give the same result on each CG. What I don't understand is whether non-identifiability in PAGs, means that there is at least a non-identifiable CG compatible with the PAG or it might be the case that all the CGs are identifiable but leading to different values. A clarification on these points would be helpful, at least from my point of view.

**Limitations:**

I don't see critical issues in terms of societal impact.

**Strengths And Weaknesses:**

Strength

- Although representing an evolution of existing results (Zhang's results and calculus for DAGs and Jaber's algorithms and completeness result) the results in this paper are clearly novel and authors' final claim about their paper "clos[ing] the problem of effect identification under Markov equivalence" is hard to question. This makes the contribution highly significant.

- The paper is very technical, but the results and the proofs seem to be correct (I only partially checked to long proofs in the supplementary material). The the derivations and the notation is quite accessible to people working with SCMs. I also appreciate the insights provided by the different examples reported in the paper.

Weaknesses

- While the statements of the theorems and the examples are generally very clear, the presentation of the algorithms can be probably improved.

---

> ### Author Response · Authors · 2022-08-02
> **Response to Reviewer CdvZ**
>
> Thank you for the very favorable assessment and the feedback provided. Below, we address the questions you raised.
>
> 1. “It is not perfectly clear to me whether or not these results can be also applied to counterfactual queries (I think so but this point might be made explicit).”
>
> Even when the specific causal diagram is available (instead of the equivalence class as in our case), counterfactual inferences require distinct machinery beyond the scope of the calculus rules and ID algorithm we discussed. One distinctive challenge with counterfactual queries in the context of PAGs is how to generalize the notion of counterfactual graphs from PAGs so as to read off relevant properties from them. The results in this paper can provide insights on how to solve this task, which is an ongoing project. We will add some remarks on this important question.
>
> 2. “As I understand, PAG identifiability corresponds to identifiability on all the CGs compatible with the PAG and the fact that the query would give the same result on each CG. What I don't understand is whether non-identifiability in PAGs, means that there is at least a non-identifiable CG compatible with the PAG or it might be the case that all the CGs are identifiable but leading to different values.”
>
> That’s a good question, thank you. According to Def. 4, non-identifiability given PAGs, in principle, can originate from one of the two conditions you mention: either the effect is not identifiable in some causal diagram in the equivalence class of the PAG, or the effect is identifiable in all the diagrams but it is not unique. Our completeness result (Thm. 4) concludes that for every non-identifiable effect given a PAG, there exists at least one causal diagram in the equivalence class of the PAG where the effect is not identifiable. This is stated in the contributions (lines 72--74) and the discussion before Thm. 4 (lines 317--323). We will make this point more prominent in the manuscript.

---

### Official Review · Reviewer_Hoya · 2022-07-11

**Rating:** 7
**Confidence:** 4
**Soundness:** 3 good
**Presentation:** 3 good
**Contribution:** 3 good

**Summary:**

This work studies the question of identification of conditional causal effects given a PAG. Authors propose sound and complete algorithm for the identification of the causal effects of the form P(y|do(x), z).


**Questions:**

It is important to provide high-level intuition about the difference between the algorithm for identifying conditional causal effects proposed in “Identification of Conditional Causal Effects under Markov Equivalence” by A. Jaber and the algorithm proposed in this work.


It will improve the readability to  add more details of what has been already done (like Algorithm 1) and what is newly proposed in the paper.

Papers on identifiability commonly have an assumption that the underlying DAG is semi-Markovian.
I would like to ask whether it is also needed here and whether this has any effect (limitation) on the set of PAGs which are considered in this work? Will the proofs still hold if the underlying DAG of the corresponding PAG was not semi-Markovian?


**Limitations:**

Please see above comments.

**Strengths And Weaknesses:**

It studies an important and quite interesting problem in causality which is causal effect identification in PAGs.

Authors propose new set of do-calculus rules for PAGs and prove their soundness and completeness in terms of identifiability of conditional causal effects. Also they propose sound and complete algorithm for the identification of causal effects P(y|do(x), z) given PAG.


There are statements that require more references. For example, the IDP algorithm is similar (except slight changes in one line)  to the algorithm in “Causal Identification under Markov Equivalence: Completeness Results” by A. Jaber. This needs to be clarified in the main text.

Results resemble the ones presented by Shpitser for identification of the conditional causal effects in DAGs. Moreover, the problem of identification of causal effects in PAGs has been addressed before. Therefore, the natural question is the contribution of this work. It is very important to clearly mention what is the main difference between the proposed method and the other work and what is the significant of this work compared to the others, e.g., proofs are different and more involved, assumptions have been relaxed, ...

---

> ### Author Response · Authors · 2022-08-02
> **Response to Reviewer Hoya (Part 2/2)**
>
> 2. “It is important to provide high-level intuition about the difference between the algorithm for identifying conditional causal effects proposed in “Identification of Conditional Causal Effects under Markov Equivalence” by A. Jaber and the algorithm proposed in this work.”
>
> The algorithm proposed in (Jaber et al., 2019b) takes a more algebraic approach that generalizes the notion of c-component in causal diagrams to pc-component in PAGs. It then exploits the functional dependencies that can be read from the PAG to rewrite and decompose the query in an effort to identify a set of sub-queries that are necessary for the identification of the input conditional effect. This approach builds on the methodology first developed in (Tian, 2004), which identifies conditional effects given a causal diagram. Most importantly, the algorithm in (Jaber et al., 2019b) was not proven to be complete, and to our knowledge, it is still unknown whether it is complete.
>
> By contrast, our algorithm follows a more symbolic approach by using the new calculus, namely rule 2, to transition from the original query to a more amenable conditional query,  where the identification of the marginal effect is necessary for the identification of the conditional. You are right to note that our approach to conditional identification resembles, in part, that of (Shpitser and Pearl, 2006), which is why we acknowledge the work in lines 274-275. However, only mimicking their treatment in PAGs falls short of identifying some conditional effects, which we illustrate explicitly in Observation 3 (lines 284--292). In general, addressing the problem of identification in PAGs obviously builds on the previous work and understanding on identification in causal diagrams; however, it comes with its own distinctive challenges such as the one discussed in Observation 3, which arise due to the structural uncertainties in PAGs.
>
> 3. “Papers on identifiability commonly have an assumption that the underlying DAG is semi-Markovian. I would like to ask whether it is also needed here and whether this has any effect (limitation) on the set of PAGs which are considered in this work? Will the proofs still hold if the underlying DAG of the corresponding PAG was not semi-Markovian?”
>
> This is an important point that we have not made explicit in the paper, so thank you for pointing it out. In fact, the reduction of CIDP to a sequence of applications of the calculus implies that the algorithm is sound and complete even when the underlying DAG is non-Markovian and not just for the special case of semi-Markovian diagrams. This result holds because both the original calculus (Zhang, 2007) and the one in Thm. 2 consider non-Markovian models. We will definitely add a note to make this point clear.

---

> ### Author Response · Authors · 2022-08-02
> **Response to Reviewer Hoya (Part 1/2)**
>
> Thank you for your assessment and time. Below, we address the raised issues.
>
> 1. “the natural question is the contribution of this work. It is very important to clearly mention what is the main difference between the proposed method and the other work and what is the significant of this work compared to the others … It will improve the readability to add more details of what has been already done (like Algorithm 1) and what is newly proposed in the paper.”
>
> Thank you for raising this issue. We share your concern about distinguishing the contributions of this work clearly from the known results in the literature. In the introduction of the manuscript, we briefly discuss the related work and its shortcomings (lines 55-65), which motivates the subsequent list of contributions (lines 69-75). We state those contributions below and contrast them with the previous work.
>
> - Introduction of a new calculus for PAGs: We propose a new calculus in Section 3.2 that subsumes that of (Zhang, 2007), and resolves its shortcomings, as discussed in Section 3.1. One subtle, yet crucial difference between the two calculi is the sufficiency of blocking the definite status paths only (Def. 3) in the new calculus in contrast to blocking all the paths in the old calculus (Prop. 1 and Thm. 1). This was posed as an open question in (Zhang, 2008a, Footnote 15) and an affirmative answer to this question is key in establishing the new calculus in Thm. 1.
>
> - Development of the first complete algorithm for conditional effect identification (Section 4.2; Algorithm 2): This algorithm employs the new calculus (Thm. 2) and Algorithm 1, a simplified version of IDP due to (Jaber et al., 2019a). Note that the original version of IDP (Jaber et al., 2019a) can be used interchangeably with Algorithm 1 in the formulation of Algorithm 2 and it still yields a complete algorithm for conditional identification. The advantage of the version in Algorithm 1 is that it outputs simpler expressions when the marginal effect is identifiable (lines 251--258). Having said that, it is important to note that we do not claim the version of IDP presented in Algorithm 1 as part of our main contributions in the introduction (lines 69-75), and hence do not intend to take any credit away from the work in (Jaber et al., 2019a).
>
> - Completeness of the proposed calculus for the task of causal effect identification (marginal and conditional): This result is obtained by reducing CIDP (Algorithm 2) to a sequence of applications of the calculus rules in Thm. 2. Hence, the completeness of CIDP implies the completeness of the new calculus for the task of identification (Thm. 4).

---

> ### Comment · Reviewer_Hoya · 2022-08-09
> **Response to author rebuttal**
>
> The authors answer most of my questions/concerns and I would like to increase my score.

---

### Official Review · Reviewer_dzjw · 2022-07-12

**Rating:** 7
**Confidence:** 2
**Soundness:** 3 good
**Presentation:** 2 fair
**Contribution:** 4 excellent

**Summary:**

This paper studies the problem of conditional causal effect identification, which asks to recover P(y|do(x), z) from the observational distribution. There is a sequence of works that propose an algorithm for this problem under the assumption that the underlining causal graph is known. This paper relaxes this assumption by assuming that only the Markov equivalence class of the underlining causal graph is known. More specifically, the paper assumes that the so-called partial ancestral graph (PAG) is known. The paper proposes new calculus, which is a modified version of Zhang's [2007] Calculus. For the new calculus, the authors prove that it is complete and atomic complete. Moreover, they propose an algorithm for the identification of conditional causal effects given a PAG and show that the algorithms is complete, that is if some causal cannot be identified by this algorithm then it is not identifiable in some causal graph in the equivalence class of a given PAG.

**Questions:**

1) What is the main difference between the algorithm proposed in by Jaber et al. 2019a and your algorithm? If the difference is insignificant, I believe the algorithm of Jaber et al. 2019a deserves more credit in the discussion, in particular, it should be acknowledged that your algorithm is a minor modification. Alternatively, if the difference is significant, I would kindly suggest including a more detailed discussion of the distinction between these algorithms.

2)  Are there any runtime or space complexity guarantees for your algorithm? Is it polynomial time?

3) Is anything known about the approximate version of the problem: how many samples are sufficient to identify the causal effect?

4) Are there any open problems left related to conditional effect identification (that authors consider interesting)?

4) I would kindly suggest renaming Proposition 1 (by Zhang 2007) into a Theorem, as calling it a proposition appears to be a bit judgemental.

5) typo in line 213: X, Y, Y

6) I believe the notation in line 265 is poorly chosen, y serves the role of both a fixed variable and a summation index (which makes no mathematical sense). I kindly suggest renaming the summation index. The same notation problem occurs in other places.

**Limitations:**

I believe the authors should comment on the runtime and sample complexity guarantees of the proposed algorithm. If the problem of establishing such guarantees is not resolved I kindly suggest that the authors acknowledge this. Alternatively, if such guarantees (lower or upper bounds) follow from the prior work it will be useful if they are included in the paper.

**Strengths And Weaknesses:**

*Significance:* The problem of identifiability of interventional causal effects is an important and long-studied problem. This paper improves over the state-of-the-art result by relaxing the assumption that the ground truth causal graph is known. Instead, the paper assumes that only the partial ancestral graph is known. This is a significant relaxation of the assumption as the true underlining causal graph is frequently not known, while PAG can be recovered from conditional independencies of variables. Zhang [2007] proposed a calculus to solve the problem considered in this paper which relies on the notion of ``possibly m-connected paths". However, Zhang's calculus is not complete, i.e., it is possible to contract a PAG in which the effect is identifiable but is not deducible from Zhang's calculus.

The key observation of this paper is that if one relies on the notion "definite m-connected paths" instead of "possibly m-connected paths", then the minor modification of Zhang's calculus becomes a complete calculus. Moreover, the paper proposes an algorithm that uses the rules of the newly introduced calculus to recover causal effect and shows that this algorithm is complete.

*Clarity:* Despite the fact the paper is reasonably well-written it is extremely technical, and as a result quite hard to read. I was not able to verify the correctness of the proofs in the appendix within a reasonable time.

*Originality:* The paper relies mostly on ideas that are well-known in the field. The proposed algorithm appears to be a minor modification of the prior algorithm by Jaber et al. 2019a. However, the proposed calculus and its completeness seem to be novel.

---

> ### Author Response · Authors · 2022-08-02
> **Response to Reviewer dzjw (Part 2/2)**
>
> 3. “Are there any open problems left related to conditional effect identification (that authors consider interesting)? … Is anything known about the approximate version of the problem: how many samples are sufficient to identify the causal effect?”
>
> We address questions Q.3 and Q.4 together as they are related. It helps to make the distinction between the task of causal effect identification and that of causal effect estimation.
>
> This work is concerned with the first task (causal identification), which asks whether a target conditional effect is uniquely computable from P(V), the observational distribution, and given a PAG learnable from P(V). The objective of CIDP in Algorithm 2 is to decide whether the effect is identifiable and provide an expression for it when the answer is yes, while being agnostic as to whether P(V) can be accurately estimated from the available samples. An interesting direction for future work is to investigate the setting where the causal effect is not identifiable given the PAG, according to Definition 4, and study whether the causal effect can be bounded (Balke and Pearl, 1997).
>
> As for the second task, the estimation of the conditional causal effect using the identification formula provided by Algorithm 2 poses several challenges under finite data. The number of samples sufficient to identify a given effect would depend on the size of the expression, among other factors, and naive methods for estimation exacerbate this problem. Recent work such as (Jung et al., 2021) proposes a double machine learning estimator for marginal effects that are identifiable given a PAG. An interesting direction of work is to generalize this approach to conditional causal effects identifiable by CIDP.
>
> 4. “I would kindly suggest renaming Proposition 1 (by Zhang 2007) into a Theorem, as calling it a proposition appears to be a bit judgemental.”
>
> Thank you for the suggestion. The choice to present this result as a proposition was not intended to diminish the groundbreaking work of (Zhang, 2007), but rather to distinguish previous work from our main contributions. We can add a clarification that propositions are from other papers while the label ‘theorem’ is reserved for the main contributions claimed in our paper. We are certainly open to reconsidering the issue in case there is an agreement among the reviewers about this convention.
>
> (Jung et al., 2021) Yonghan Jung, Jin Tian, and Elias Bareinboim. "Estimating identifiable causal effects on Markov equivalence class through double machine learning." International Conference on Machine Learning. PMLR, 2021.
>
> (van der Zander et al., 2019) Benito van der Zander, Maciej Liśkiewicz, and Johannes Textor. “Separators and adjustment sets in causal graphs: Complete criteria and an algorithmic framework.” Artificial Intelligence 270 (2019): 1-40.
>
> (Shacter, 1998) Ross D. Shacter. "Bayes ball: The rational pastime." In Proceedings of the 14 Annual Conference on Uncertainty in Artificial Intelligence. 1998.
>
> (Balke and Pearl, 1997) Alexander Balke and Judea Pearl. "Bounds on treatment effects from studies with imperfect compliance." Journal of the American Statistical Association 92, no. 439 (1997): 1171-1176.

---

> > ### Comment · Reviewer_dzjw · 2022-08-09
> > **Thank you for the reply!**
> >
> > I would like to thank the authors of the paper for their detailed reply to my questions!

---

> ### Author Response · Authors · 2022-08-02
> **Response to Reviewer dzjw (Part 1/2)**
>
> Thank you for the favorable assessment of our work and all the feedback provided. We will fix the typos and notations accordingly. Below we address the other raised issues:
>
> 1. “What is the main difference between the algorithm proposed in by Jaber et al. 2019a and your algorithm? If the difference is insignificant, I believe the algorithm of Jaber et al. 2019a deserves more credit in the discussion…”
>
> Thank you for raising this issue. We have no intention to take any credit away from the work of Jaber et al. (2019a), where the first complete algorithm for marginal effect identification was presented. Notice that we don't claim Algorithm 1 (the presented version of IDP) as part of the contributions, which are listed at the end of the introduction on page 2. Moreover, we introduce the corresponding section on marginal effect identification by saying “Sec. 4.1 introduces a version of the IDP algorithm [Jaber et al., 2019a] to identify marginal causal effects. The attractiveness of this version is that it yields simpler expressions whenever the effect is identifiable while preserving the same expressive power, i.e., completeness for marginal identification.” In that sense, Algorithm 1 is a mere simplification of the algorithm proposed in (Jaber et al., 2019a), which abstracts away some details and is hopefully easier to understand.
>
> On the other hand, we do claim CIDP (Algorithm 2) as the second main contribution, in addition to the proposed calculus. CIDP transforms a given conditional query to a specific, and more amenable conditional query using rule 2 of the proposed calculus before calling IDP as a subroutine to identify the marginal effect if possible. However, the use of rule 2 of the calculus to obtain this reduction is quite subtle and non-trivial, as discussed in Section 4.2 (based on the three observations therein). Last but not least, it is important to note that the completeness of Algorithm 2 is the reason why we are able to establish the completeness of the calculus rules for the task of causal effect identification. So, while the atomic completeness of the calculus (Thm. 2) is a standalone consequence of the calculus itself, the completeness of the calculus for the task of causal identification (Thm. 4) is derived from the completeness of the CIDP algorithm.
>
> 2. “Are there any runtime or space complexity guarantees for your algorithm? Is it polynomial time?”
>
> Similar to the ID and CID algorithms for DAGs (Tian & Pearl, 2002; Tian, 2004), the IDP and CIDP algorithms run in polynomial time in the number of nodes in the graph. Operations for finding possible ancestors and descendants as well as finding buckets and pc-components of a node (lines 1 and 6 of IDP and lines 1, 2, and 6 of CIDP) are performed by graph traversal (e.g., depth-first search). Each of these operations is linear in the number of nodes in the graph. Specifically, each search takes time O(n + m) where n is the number of nodes and m is the number of edges. Testing definite m-separability in PAGs can also be done in time O(n + m) by a procedure based on the Bayes-Ball algorithm (Shachter, 1998). A similar procedure has been used by (Perkovic et al., 2018) for deciding m-separability when testing whether a set is admissible for adjustment in PAGs. The complexity is the same for deciding m-separability in MAGs, which was shown to be O(n + m) by (Van der Zander et al., 2019). Since applications of do-calculus in PAGs rely on testing m-separation, lines 4 and 8 of CIDP run in O(n + m). Finally, the partition process in line 10 of IDP is clearly polynomial.

---

### Meta-Review · Area_Chair_imje · 2022-08-24

**Recommendation:** Accept
**Confidence:** Certain

**Metareview:**

The reviewers are all in agreeement that the paper constitutes a fundamental advance in the theory of causal inference. The authors responded to the reviewers' remaining questions in a detailed way, and there is no further issue with the paper being accepted.

**Award:**

Yes

---

### Decision · Program_Chairs · 2022-09-14

Accept